# Computation Screening of Multi-Target Antidiabetic Properties of Phytochemicals in Common Edible Mediterranean Plants

**DOI:** 10.3390/plants11131637

**Published:** 2022-06-21

**Authors:** Vlasios Goulas, Antonio J. Banegas-Luna, Athena Constantinou, Horacio Pérez-Sánchez, Alexandra Barbouti

**Affiliations:** 1Department of Agricultural Sciences, Biotechnology and Food Science, Cyprus University of Technology, Lemesos 3603, Cyprus; ai.konstantinou@edu.cut.ac.cy; 2Structural Bioinformatics and High Performance Computing (BIO-HPC) Research Group, UCAM Universidad Católica de Murcia, 30107 Guadalupe, Spain; ajbanegas@ucam.edu (A.J.B.-L.); hperez@ucam.edu (H.P.-S.); 3Department of Anatomy-Histology-Embryology, Faculty of Medicine, School of Health Sciences, University of Ioannina, 45110 Ioannina, Greece; abarbout@uoi.gr

**Keywords:** antidiabetic activity, diabetes, DIA-DB web server, flavonoids, in silico study, mechanism of action, phenolic acids, terpenes

## Abstract

Diabetes mellitus is a metabolic disease and one of the leading causes of deaths worldwide. Numerous studies support that the Mediterranean diet has preventive and treatment effects on diabetes. These effects have been attributed to the special bioactive composition of Mediterranean foods. The objective of this work was to decipher the antidiabetic activity of Mediterranean edible plant materials using the DIA-DB inverse virtual screening web server. A literature review on the antidiabetic potential of Mediterranean plants was performed and twenty plants were selected for further examination. Subsequently, the most abundant flavonoids, phenolic acids, and terpenes in plant materials were studied to predict their antidiabetic activity. Results showed that flavonoids are the most active phytochemicals as they modulate the function of 17 protein-targets and present high structural similarity with antidiabetic drugs. Their antidiabetic effects are linked with three mechanisms of action, namely (i) regulation of insulin secretion/sensitivity, (ii) regulation of glucose metabolism, and (iii) regulation of lipid metabolism. Overall, the findings can be utilized to understand the antidiabetic activity of edible Mediterranean plants pinpointing the most active phytoconstituents.

## 1. Introduction

Diabetes mellitus (DM), or merely diabetes, is a complex chronic disease, which requires continues medical care, reducing the patient’s quality of life and raising the medical cost if not treated properly. According to the World Health Organization (WHO), DM has entered the top 10 leading causes of deaths worldwide, following a significant percentage increase of 70% from 2000 to 2030 [1]. From a medical point of view, it is a chronic noncommunicable disease arising from impaired insulin secretion and insulin resistance, leading to its defining feature of hyperglycemia. It can affect different organ systems in the body and, over time, causes several complications including neuropathy, nephropathy, retinopathy, cardiovascular disease, stroke, and peripheral artery disease [2]. The more common types of DM are classified into type 1 (diabetes mellitus (T1DM)), type 2, and gestational diabetes. Type 1 DM is characterized by absolute insulin deficiency associated with pancreatic β cells destruction [3], while type 2 DM, which accounts for nearly 95% of individuals, is mainly due to insulin resistance (IR) and deficiency in insulin secretion. Gestational diabetes, on the other hand, develops during pregnancy and usually disappears after giving birth. [3].

Recently, Newman and Cragg (2019) stated and classified all approved therapeutic agents for DM, from the 1st of January 1981 to the 30th of September 2019. According to their report, over 54% of registered drugs for DM are natural products or mimic natural products. It is obvious that natural products are an attractive reservoir of antidiabetic compounds [4]. In addition, numerous studies demonstrate an antidiabetic potential of natural products [5]. The antidiabetic effects of many phytochemicals including polyphenols, terpenes, alkaloids, saponins, and quinones have been well-documented [6,7]. Furthermore, clinical trials with medicinal plants and natural products have been conducted, whereas some of them have been used for the development of herbal formulations controlling DM. Regarding the mechanism of action of natural products, (i) the inhibition of α-glucosidase and α-amylase in the digestive tract, (ii) the boost of insulin secretion and pancreatic β cell proliferation, (iii) the regulation of glucose uptake and glucose transporters, (iv) the inhibition of protein tyrosine phosphatase 1B activity, and (v) the reduction in the generation of oxidative stress are the main modes of action of pure phytochemicals and crude extracts. [3].

Epidemiological studies also highlight that the adoption of a healthy dietary pattern such as the Mediterranean diet contributes to the prevention and treatment of Type 2 DM [8,9]. The beneficial effects of the Mediterranean diet are correlated with weight control and the consumption of foods rich in nutrients with various health benefits [10]. Fruit and vegetables are essential components of the Mediterranean diet and contain many bioactive phytochemicals. Taking into consideration a current study where researchers fished new antidiabetic compounds from various natural products, the search for new antidiabetic agents in Mediterranean edible plants stands as a challenge [5].

In the present study, we strived to clarify the antidiabetic potential of edible Mediterranean plant materials that contain common phytochemicals, with the employment of in silico virtual screening methodologies. Plants are complex mixtures of several primary and secondary metabolites and the classic approach including the isolation and the evaluation of the antidiabetic activity is time-consuming and complicated. Thus, the employment of the DIA-DB inverse virtual screening web server allows the rapid evaluation of several compounds for antidiabetic activity and predicts the possible mode of action of antidiabetic compounds.

## 2. Results and Discussion

### 2.1. Plant and Phytochemical Selection

At first, a list of plants that are widely distributed in Mediterranean flora and/or the Mediterranean diet was prepared. The in vitro and in vivo antidiabetic activity of these plants were checked with the implementation of an extensive literature review. Twenty plants were selected for further examination as they exert both in vitro and in vivo antidiabetic activity (Table 1). All selected plants are edible and/or can be consumed after cooking or processing. Regarding the plant taxonomy, they belong to 11 families. Five plants come from the Lamiaceae and four from the Apiaceae family. Both Amaryllidaceae and Asteraceae families includes two plant materials for further study. Other plant materials belong to Brassicaceae, Ranunculaceae, Plantaginaceae, Rutaceae, Rosaceae, Oleaceaceae, and Vitaceae families.

A literature review showed that the in vitro antidiabetic activity of Mediterranean plants is mostly correlated with their inhibitory effects on enzymes related to DM. About 85% of the selected plant material have α-glucosidase and/or α-amylase inhibitory activity. Both enzymes are considered as carbohydrate-hydrolyzing enzymes and are linked with postprandial hyperglycaemia as they regulate the absorption of glucose [11]. Studies also demonstrated the stimulation of insulin secretion using different pancreatic β-cells [12,13]. Coriander and black mustard seeds, thyme, and summer savory also enhance in vitro the glucose uptake in cell lines or rat muscle pieces [12,13,14]. Furthermore, in vitro studies indicated that Mediterranean plants promote the proliferation of pancreatic β-cells [13], inhibit dipeptidyl peptidase-4 inhibition [15], prevent the formation of advanced glycation end-products [16], and decrease the fat accumulation in *Caenorhabditis elegans* [14].

Several works also document the in vivo antidiabetic effects of selected Mediterranean plants (Table 1). This activity is mainly established using animal models such as healthy and streptozotocin or alloxan-induced diabetic rats or mice. There are also clinical trials that report their antidiabetic activity in patients with type 2 DM and overweight adults. The reduction in glucose in blood, the stimulation of insulin secretion, and the antihyperlipidemic properties are the main antidiabetic effects of selected Mediterranean plants according to the literature review. In addition, the decrease in glycosylated hemoglobin (HbA1c), the protective effects from tissue damage, and the improvement of liver functions have been also mentioned. It is noteworthy that all selected plants exert antidiabetic effects through multiple mechanisms. It is attributed to the synergism of two or more phytochemicals as plants are complex mixtures of bioactive phytochemicals.

In the next step, the phytochemical composition of active Mediterranean plants was elucidated. The constituents were classified into three groups, namely phenolic acids, flavonoids, and terpenes. The most abundant phytochemicals from each group were selected to further investigate their antidiabetic potential using the DIA-DB inverse virtual screening web server. Table 2 summarizes the phytochemicals and their distribution in the Mediterranean plants, which is as follows: quercetin is present in 19 plants, caffeic acid in 16, ferulic acid in 15, and luteolin in 15 plants. These are the most common phytochemicals in the Mediterranean plants in the present study. On the other hand, chrysoeriol, ellagic acid, and hesperidin are found less frequently in Mediterranean plants (in 5, 6, and 6 plants, respectively). All structures of the studied phytoconstituents are presented in Figure 1.

### 2.2. Estimation of Antidiabetic Activity of Phytochemicals using Virtual Screening

The phytochemicals were screened for antidiabetic activity against 17 diabetes targets using the DIA-DB inverse virtual screening web server. The docking scores of the crystallized ligands ranged from −10.4 to −1.5 kcal mol^−1^. A docking cutoff score of −8 kcal mol^−1^ was set, as it was deemed a reasonable average docking score that covered the most active compounds for each protein target (Table 3). Results revealed interesting findings for the antidiabetic potential of phytochemicals. More specifically, all terpenes presented docking scores bigger than −8 kcal mol^−1^; thus, the predicted antidiabetic activities of all terpenes are negligible or weak. The studied terpenes cannot be considered promising antidiabetic compounds for further investigation. In addition, the antidiabetic effects of edible plant materials cannot be correlated with the presence of these terpenes.

Ellagic, caffeic, and ferulic acids were active phenolic acids (Table 3). The antidiabetic potential of ellagic acid is linked with five protein targets, whereas the regulation, secretion, and/or sensitivity of insulin is the main mechanism of action of ellagic acid. In vitro and in vivo studies also reported the antidiabetic effect of ellagic acid through the stimulation of insulin [165,166]. Regarding hydroxycinnamic acids, caffeic and ferulic acids modulate the functions of only one protein target. Previous studies also manifest their antidiabetic effects by modulating insulin-signaling molecules [167,168]. A quantitative structure–activity relationship analysis for hydroxycinnamic acids will be interesting as the other acids such cinnamic acid, coumaric acid, and rosmarinic acid did not bInd. with protein targets.

Indisputably, results demonstrated that flavonoids are the most promising group of compounds. All studied flavonoids bInd. with protein targets related with DM. The antidiabetic potential of flavonoids is correlated with three possible mechanisms, namely the (i) regulation of insulin secretion and/or sensitivity, (ii) regulation of glucose metabolism, and (iii) regulation of lipid metabolism. DPP4, HSD11B1, AKR1B1, AMY2A, and PPARG protein targets interacted with the majority of flavonoids. Figure 2 shows hesperidin and rutin binds with 15 and 12 protein targets, respectively, whereas the other flavonoids modulate the function of 6 to 8 protein targets. It is noteworthy that both hesperidin and rutin are glycosylated flavonoids. Furthermore, a comparison between rutin and its aglycone form (quercetin) demonstrates the superiority of the glycosylated form. Jadhav and Puchchakayala (2012) also stated that rutin had a higher antidiabetic and hypoglycemic activity than quercetin in normoglycemic and streptozotocin -nicotinamide-induced diabetic rats performing blood glucose and serum lipid profile measurements [169]. Hesperidin is less distributed than rutin but is also a strong multi-target antidiabetic agent according to the computational measurements. These findings are in line with the literature; hesperidin is considered to have preventive and/or therapeutic effects against DM as it regulates glucose and lipid metabolism [170,171]. Overall, flavonoids are an interesting pool of antidiabetic compounds and further study is needed to probe the structural characteristics of flavonoids that are mainly linked with antidiabetic activity.

### 2.3. Evaluation of Molecular Similarity of Predicted Active Phytochemicals and Known/Experimental Antidiabetic Drugs

In the next step, the Tanimoto similarity index was used to quantify the similarity of the studied phytochemicals with 190 known or experimental antidiabetic drugs. A Tanimoto score of 0.7 or greater indicated a robust molecular similarity. According to our results, all tested phytochemicals have structural similarities with antidiabetic drugs. Although these findings are contrary to their predicted antidiabetic activity, the in-depth interpretation of results confirmed the previous results. More specifically, the highest molecular similarity was found for flavonoids. A previous study also correlated flavonoids with the antidiabetic activity of herbs and spices [2]. A strong molecular similarity with 132 and 131 antidiabetic drugs was detected for hesperidin (69.5%) and rutin (68.9%). Both flavonoids also bInd. with the highest number of protein targets according to our results. Furthermore, chrysoeriol (57.4%), luteolin (51.1%), and apigenin (50.5%) also showed some structural similarity with antidiabetic drugs. Similarly to the predicted antidiabetic activity, the glycosylated flavonoids had more structural similarities with antidiabetic drugs than aglycone flavonoids. From a chemical point of view, the most active flavonoids belong to the group of flavanones, flavanols, and flavones. In summary, the molecular similarity test also confirmed the potent antidiabetic effects of flavonoids.

## 3. Materials and Methods

### 3.1. Literature Review

The literature review on the antidiabetic potential of Mediterranean plants was performed using the databases of Scopus [172] and Google Scholar [173]. More specifically, the search included the terms “antidiabetic activity”, “diabetes”, “hyperglycemia”, “hypoglycemic activity”, “alpha-glucosidase”, and “alpha-amylase” in combination with “scientific plant name” and “common plant name” of Mediterranean plants. The antidiabetic potency of several plants was searched according to a list, which was prepared based on their abundance in the Mediterranean flora and/or diet.

Subsequently, data for the phytochemical composition of plants were collected using the same databases. The terms “scientific plant name”, “common plant name”, “phytochemicals”, “bioactive compounds”, “Liquid chromatography”, “Nuclear Magnetic Resonance (NMR)”, and “Liquid Chromatography-Mass Spectroscopy (LC-MS)” were used to build the compound library.

### 3.2. Determination of Anti-Diabetic Activity Using DIA-DB Inverse Virtual Screening Web Server

At first, a simplified molecular-input line-entry system (SMILES) notation was created for each compound, which was obtained from PubChem. [174]. All SMILES notations are demonstrated in Table 3. The SMILES notation of each compound was subsequently submitted to the DIA-DB web server that employs an inverse virtual screening of compounds with Autodock Vina against a given set of 18 protein targets associated with diabetes. The 18 protein targets can be classified into the three following groups: (i) regulation of insulin secretion and/or sensitivity (DPP4, FFAR1, HSD11B1, INSR, PTPN9, RBP4), (ii) regulation of glucose metabolism (AKR1B1, AMY2A, FBP1, GCK, MGAM, PDK2, PYGL), and (iii) regulation of lipid metabolism (NR5A2, PPARA, PPARD, PPARG, RXRA) [2,175]. More specifically, the targets were aldose reductase (AKR1B1), AMY2A, FBP1, free fatty acid receptor 1 (FFAR1), glucokinase (GCK), 11B-hydroxysteroid dehydrogenase type 1 (HSD11B1), insulin receptor (INSR), MGAM, NR5A2, pyruvate dehydrogenase kinase isoform 2 (PDK2), PPARA, PPARD, PPARG, PTPN9, liver glycogen phosphorylase (PYGL), RBP4, and retinoid X receptor alpha (RXRA). A cut-off docking score of −8 kcal mol^−1^ was set to distinguish between potential active and inactive compounds. 

### 3.3. Evaluation of Molecular Similarity of Predicted Active Phytochemicals and Known/Experimental Antidiabetic Drugs

The similarities studies with known/experimental antidiabetic drugs were performed according to a previous work [176]. The molecular similarity was performed using the metric of the Tanimoto similarity on the calculated ECFP4 molecular fingerprints of the compounds. The molecular similarity network was generated with Cytoscape and the ChemViz2 Application version 1.1.0.

## 4. Conclusions

The present study was undertaken to shed light on the antidiabetic properties of edible Mediterranean plants. Our results showed that the most active compounds within examined Mediterranean plants are flavonoids that are widely distributed in herbs, vegetables, medicinal plants, and fruits. Our findings also show that the glycosylation of flavonoids potentially improves its antidiabetic activity. Both the evaluation of antidiabetic activity and molecular similarity of antidiabetic dugs highlighted the antidiabetic potential of hesperidin and rutin. Finally, the present study showed that the employment of the DIA-DB inverse virtual screening web server allows us to explain the antidiabetic effects of natural products and to pinpoint the most active compounds.

## Figures and Tables

**Figure 1 plants-11-01637-f001:**
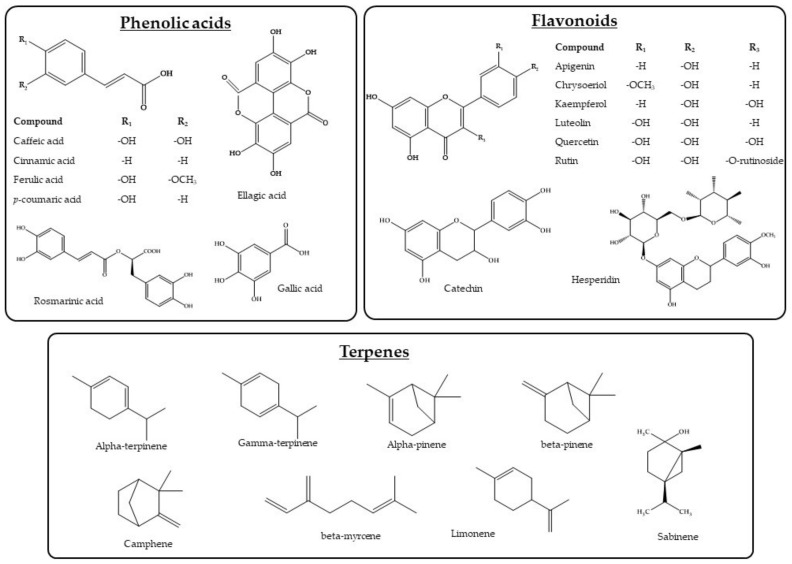
Structures of the studied phytochemicals present in Mediterranean plants.

**Figure 2 plants-11-01637-f002:**
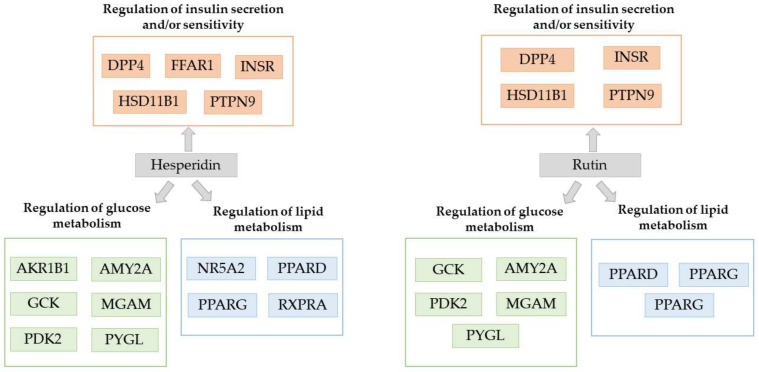
Interaction of hesperidin and rutin with protein targets related with diabetes mellitus as predicted with the employment of DIA-DB inverse virtual screening web server.

**Table 1 plants-11-01637-t001:** A comprehensive summary of in vitro and in vivo antidiabetic properties of edible Mediterranean plants.

Common Name	Scientific Name	Plant Part	In vitro Antidiabetic Effects	In vivo Antidiabetic Effects	References
Black cumin	*Nigella sativa* L.	Seeds	▪ Increase in insulin secretion ▪ Induction in proliferation of pancreatic β-cells ▪ Stimulation of glucose uptake	▪ Reduction in blood glucose ▪ Antihyperlipidemic effects	[13,17,18,19]
Black mustard	*Brassica nigra* L.	Aerial plants, seeds	▪ α-glucosidase inhibition	▪ Reduction in blood glucose ▪ Decrease in glycosylated hemoglobin (HbA1c) ▪ Antihyperlipidemic effects ▪ Insulinotropic effect	[20,21,22]
Broadleaf plantain	*Plantago major* L.	Leaves	▪ α-amylase inhibition	▪ Reduction in blood glucose ▪ Stimulation of insulin secretion	[23,24,25]
Citrus fruits	Citrus spp.	Peel	▪ α-glucosidase inhibition ▪ α-amylase inhibition	▪ Reduction in blood glucose ▪ Antihyperlipidemic effects ▪ Stimulation of insulin secretion ▪ Protective effects for tissue damage	[26,27,28,29,30]
Coriander	*Coriandrum sativum* L.	Seeds	▪ Stimulation of insulin secretion ▪ Enhancement of glucose uptake	▪ Stimulation of insulin secretion ▪ Enhancement of glucose uptake ▪ Reduction in blood glucose ▪ Antihyperlipidemic effects	[12]
Cumin	*Cuminum cyminum* L.	Seeds	▪ α-glucosidase inhibition ▪ α-amylase inhibition	▪ Reduction in blood glucose ▪ Antihyperlipidemic effects ▪ Decrease of HbA1c	[31,32,33,34]
Dill	*Anethum graveolens* L.	Seeds	▪ α-glucosidase inhibition ▪ α-amylase inhibition	▪ Reduction in blood glucose ▪ Antihyperlipidemic effects ▪ Control of colonic motility disorder	[35,36,37]
Garlic	*Allium sativum* L	Bulb	▪ dipeptidyl peptidase-4 inhibition ▪ α-glucosidase inhibition	▪ Reduction in blood glucose ▪ Antihyperlipidemic effects ▪ Decrease of insulin resistance	[15,38,39,40]
Grapes	*Vitis vinifera* L.	Seeds, skins	α-glucosidase inhibition α-amylase inhibition	▪ Reduction in blood glucose ▪ Antihyperlipidemic effects ▪ Decrease in HbA1c	[41,42]
Marjoram	*Origanum majorana* L.	Aerial parts	▪ α-glucosidase inhibition ▪ Inhibition of advanced glycation end-product (AGE) formation	▪ Inhibition of (AGE) formation ▪ Reduction in blood glucose ▪ Antihyperlipidemic effects	[16,43]
Olive	*Olea europaea* L.	Leaves	▪ α-glucosidase inhibition ▪ α-amylase inhibition	▪ Reduction in blood glucose ▪ Stimulation of insulin secretion ▪ Antihyperlipidemic effects ▪ Decrease in histopathological changes ▪ α-amylase inhibition	[44,45,46,47,48]
Onion	*Allium sepa* L.	Bulbs, skins	▪ α-glucosidase inhibition	▪ Reduction in blood glucose ▪ Stimulation of insulin secretion	[49,50,51,52]
Parsley	*Petroselinum sativum/crispum*	Leaves	▪ α-glucosidase inhibition ▪ α-amylase inhibition	▪ Reduction in blood glucose ▪ Stimulation of insulin secretion ▪ Antihyperlipidemic effects	[53,54,55]
Rosemary	*Rosmarinus officinalis* L.	Aerial parts	▪ α-glucosidase inhibition	▪ Stimulation of insulin secretion	[56]
Sage	*Salvia officinalis* L.	Aerial parts	▪ α-glucosidase inhibition ▪ α-amylase inhibition	▪ Reduction in blood glucose ▪ Stimulation of insulin secretion ▪ Antihyperlipidemic effects	[57,58,59]
Sow thistle	*Sonchus oleraceus* L.		▪ α-glucosidase inhibition ▪ α-amylase inhibition	▪ Reduction in blood glucose ▪ Stimulation of insulin secretion ▪ Antihyperlipidemic effects ▪ Protective effects for tissue damage	[60,61,62,63]
Strawberry	Fragaria spp.	Leaves, fruits	▪ α-glucosidase inhibition ▪ α-amylase inhibition	▪ Reduction in blood glucose ▪ Antihyperlipidemic effects Improvement of liver functions	[64,65,66,67]
Summer savory	*Satureja hortensis* L.	Aerial parts	▪ Stimulation of insulin-dependent glucose uptake	▪ Reduction in blood glucose	[68]
Thyme	*Thymus vulgaris* L.	Aerial parts	▪ α-glucosidase inhibition ▪ α-amylase inhibition ▪ Stimulation of insulin-dependent glucose uptake ▪ Decrease of fat accumulation in *Caenorhabditis elegans*	▪ Reduction in blood glucose ▪ Antihyperlipidemic effects	[14,69,70,71]
Yarrow	*Achillea santolina* L.	Aerial parts	▪ α-amylase inhibition	▪ Reduction in blood glucose ▪ Antihyperlipidemic effects	[72,73,74]

**Table 2 plants-11-01637-t002:** Occurrence of phytochemicals in Mediterranean plant materials and their SMILES notations.

Compound	SMILES Notation	Plants	References
Phenolic acids			
Caffeic acid	C1=CC(=C(C=C1C=CC(=O)O)O)O	black mustard, broadleaf plantain, citrus peel, coriander, cumin, dill, garlic, grape skin, marjoram, olive leaf, parsley, rosemary, sage, santolina, thyme	[43,75,76,77,78,79,80,81,82,83,84,85]
Cinnamic Acid	C1=CC=C(C=C1)C=CC(=O)O	dill, grape skin, marjoram, olive leaf, sage, thyme	[43,81,82,83,86,87]
Ellagic acid	C1=C2C3=C(C(=C1O)O)OC(=O)C4=CC(=C(C(=C43)OC2=O)O)O	black mustard, broadleaf plantain, citrus fruits, grape skin, parsley, strawberry	[75,88,89,90,91,92]
Ferulic acid	COC1=C(C=CC(=C1)C=CC(=O)O)O	black cumin, citrus peel, coriander, cumin, dill, garlic, marjoram, olive leaf, onion, rosemary, sage, santolina, thyme, strawberry	[43,77,78,79,82,85,87,93,94,95,96]
Gallic acid	C1=C(C=C(C(=C1O)O)O)C(=O)O	black mustard, black cumin, citrus peel, cumin, dill, garlic, grape skin and seeds, marjoram, olive leaf, onion, strawberry,	[43,76,77,82,86,93,94,97,98,99,100,101]
p-Coumaric acid	C1=CC(=CC=C1C=CC(=O)O)O	black cumin, broadleaf plantain, citrus peel, coriander, cumin, dill, garlic, grape skin, marjoram, olive leaf, parsley, sage, santolina, thyme	[43,76,77,78,79,80,81,82,84,85,87,93,99,102]
Rosmarinic acid	C1=CC(=C(C=C1CC(C(=O)O)OC(=O)C=CC2=CC(=C(C=C2)O)O)O)O	cumin, dill marjoram, olive leaf, rosemary, sage, santolina, summer savory, thyme	[43,82,83,97,102,103,104]
Flavonoids			
Apigenin	C1=CC(=CC=C1C2=CC(=O)C3=C(C=C(C=C3O2)O)O)O	black mustard, black cumin, broadleaf plantain, citrus peel, cumin, garlic, marjoram, olive leaf, parsley, santolina, sage, sow thistle, summer savory, thyme,	[43,75,76,80,82,85,87,93,97,104,105,106,107]
Catechin	C1C(C(OC2=CC(=CC(=C21)O)O)C3=CC(=C(C=C3)O)O)O	black mustard, black cumin, broadleaf plantain, citrus peel, coriander, cumin, grape skin, olive leaf, onion, strawberry	[77,78,82,93,100,108,109,110,111]
Chrysoeriol	COC1=C(C=CC(=C1)C2=CC(=O)C3=C(C=C(C=C3O2)O)O)O	broadleaf plantain, citrus peel, coriander, olive leaf, parsley,	[112,113,114,115,116]
Hesperidin	CC1C(C(C(C(O1)OCC2CC(CC(O2)OC3=CC(=C4C(=O)CC(OC4=C3)C5=CC(=C(C=C5)OΨ)O)O)O)O)O)O)O)O	citrus peels, marjoram, olive leaf, rosemary, strawberry, summer savory	[43,77,82,117,118,119]
Kaempferol	C1=CC(=CC=C1C2=C(C(=O)C3=C(C=C(C=C3O2)O)O)O)O	black mustard, citrus peel, coriander, cumin, grape skin, onion, parsley, rosemary, santolina, strawberry, summer savory, thyme	[75,77,78,85,94,99,101,120,121,122,123]
Luteolin	C1=CC(=C(C=C1C2=CC(=O)C3=C(C=C(C=C3O2)O)O)O)O	broadleaf plantain, citrus peel, coriander, cumin, garlic, marjoram, olive leaf, parsley, rosemary, sage, santolina, sow thistle, strawberry, summer savory, thyme	[123,124,125,126,127]
Quercetin	C1=CC(=C(C=C1C2=C(C(=O)C3=C(C=C(C=C3O2)O)O)O)O)O	black mustard, broadleaf plantain, citrus peel, coriander, cumin, dill, garlic, grape skin, marjoram, olive leaf, onion, parsley, rosemary, sage, sow thistle, summer savory, strawberry, thyme	[85,92,93,95,96,97,98,102,109,110,117,122,124,125,126,127,128,129]
Rutin	CC1C(C(C(C(O1)OCC2C(C(C(C(O2)OC3=C(OC4=CC(=CC(=C4C3=O)O)O)C5=CC(=C(C=C5)O)O)O)O)O)O)O)O	black mustard, broadleaf plantain, citrus peel, dill, garlic, marjoram, olive leaf, rosemary, sage, santolina, strawberry, summer savory,	[90,91,92,97,98,101,102,111,114,122,129,130]
Terpenes			
alpha-pinene	CC1=CCC2CC1C2(C)C	black mustard, citrus peel, coriander, cumin, dill, grape skins, marjoram, olive leaf, rosemary, sage, santolina, summer savory, thyme	[43,105,131,132,133,134,135,136,137,138,139]
alpha-terpinene	CC1=CC=C(CC1)C(C)C	cumin, marjoram, parsley, rosemary, sage, santolina, summer savory, thyme	[43,140,141,142,143,144,145,146]
beta-pinene	CC1(C2CCC(=C)C1C2)C	citrus peel, coriander, cumin, grape skins, marjoram, olive leaf, rosemary, sage, summer savory, thyme	[43,77,133,134,136,137,138,139]
camphene	CC1(C2CCC(C2)C1=C)C	coriander, marjoram, rosemary, sage, santolina, summer savory, thyme,	[43,74,133,138,139]
gamma-terpinene	CC1=CCC(=CC1)C(C)C	citrus peel, coriander, dill, marjoram, parsley, rosemary, sage, santolina, summer savory, thyme	[43,77,135,139,141,142,143,144,145,146,147]
limonene	CC1=CCC(CC1)C(=C)C	black mustard, citrus peel, coriander, dill, marjoram, onion, rosemary, santolina, summer savory, thyme	[43,77,132,133,135,139,142,145,146]
beta-myrcene	CC(=CCCC(=C)C=C)C	citrus peel, coriander, dill, grape skins, marjoram, olive leaf, parsley, rosemary, sage, summer savory, thyme	[43,105,133,135,136,137,138,139,141,145]
sabinene	CC(C)C12CCC(=C)C1C2	citrus peel, coriander, marjoram, olive leaf, parsley, sage, santolina, summer savory, thyme	[43,74,105,133,137,139,141,143,145]

**Table 3 plants-11-01637-t003:** The docking cut-off score of active phytochemicals for each protein target. The score is given in parentheses and expressed as kcal mol^−1^.

Protein Target	PDB Code	Function	Phytochemicals
Regulation of insulin secretion and/or sensitivity			
DPP4	4A5S	Stimulation of insulin secretion from pancreas degrading and inactivating glucagon-like peptide-1 [148]	Apigenin (−8.2), catechin (−8.3), chrysoeriol (−8.1), ellagic acid (−8.3), hesperidin (−10.4), kaempferol (−8.4), quercetin (−8.3), rutin (−9.1)
FFAR1	4PHU	Binding of free fatty acids to receptor results in increase in glucose-stimulated insulin secretion [149]	Apigenin (−8.4), caffeic acid (−8.0), ferulic acid (−8.0), hesperidin (−8.7), luteolin (−8.2)
HSD11B1	4K1L	Activates the synthesis of active glucocorticoids [150]	Apigenin (−9.0), catechin (−9.0), chrysoeriol (−9.1), ellagic acid (−8.6), hesperidin (−9.9), kaempferol (−9.2), luteolin (−9.4), quercetin (−9.8), rutin (−9.7)
INSR	3EKN	Regulates glucose uptake and synthesis of glycogen, lipid, and protein [151]	Ellagic acid (−8.2), hesperidin (−9.3), rutin (−8.4)
PTPN9	4GE6	Reduces insulin sensitivity, dephosphorylating the insulin receptor [152]	Ellagic acid (−8.0), hesperidin (−8.8), rutin (−8.6)
RBP4	2WR6	Reduces insulin signaling and promotes gluconeogenesis [153]	Apigenin (−9.9), catechin (−9.0), chrysoeriol (−9.6), ellagic acid (−8.7), kaempferol (−9.5), luteolin (−9.9), quercetin (−9.6)
Regulation of glucose metabolism			
AKR1B1	3G5E	Catalyzes the reduction of glucose to sorbitol [154]	Apigenin (−9.1), catechin (−9.1), chrysoeriol (−9.0), ellagic acid (−8.8), hesperidin (−8.1), kaempferol (−8.6), luteolin (−9.1), quercetin (−8.8)
AMY2A	4GQR	Regulates the digestion of starch to glucose [155]	Apigenin (−8.3), catechin (−8.4), chrysoeriol (−8.5), hesperidin (−8.9), kaempferol (−8.0), luteolin (−9.4), quercetin (−9.8), rutin (−9.0)
GCK	3IMX	Phosphorylates glucose for glycolysis or synthesis of glycogen [156]	Hesperidin (−10.2), rutin (−8.6)
MGAM	3L4Y	Regulates the digestion of starch to glucose [157]	Hesperidin (−8.2), rutin (−8.3)
PDK2	4MPC	Regulates glucose oxidation through the inactivation of pyruvate dehydrogenase complex [158]	Hesperidin (−9.1), rutin (−8.1)
PYGL	3DDS	Regulates phosphorolysis of glycogen in glycogenesis [159]	Hesperidin (−8.6), rutin (−8.6)
Regulation of lipid metabolism			
NR5A2	4DOR	Regulates the expression of genes involved in the synthesis of bile acid and cholesterol, and steroidogenesis [160]	Hesperidin (−8.4)
PPARA	3FEI	Regulates the expression of genes involved in lipid metabolism [161]	Rutin (−8.4)
PPARD	3PEQ	Regulates the expression of genes involved in fatty acid catabolism [162]	Chrysoeriol (−8.0), hesperidin (−9.1), rutin (−8.9)
PPARG	2FVJ	Regulates the expression of genes involved in adipogenesis and lipid oxidation [163]	Apigenin (−8.1), catechin (−8.2), chrysoeriol (−8.4), ellagic acid (−8.3), hesperidin (−1035), kaempferol (−8.5), luteolin (−8.2), quercetin (−8.4), rutin (−9.8)
RXPRA	1FM9	Heterodimerizes with PPARs [164]	Apigenin (−9.2), chrysoeriol (−9.3), hesperidin (−8.0), luteolin (−9.1)

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
