# Peer review of "Computation Screening of Multi-Target Antidiabetic Properties of Phytochemicals in Common Edible Mediterranean Plants"

_plants, 2022, doi:10.3390/plants11131637_

Round 1

Reviewer 1 Report

The article concerns the explanation of the antidiabetic action of Mediterranean edible plant materials using DIA-DB 20 inverse virtual screening web server. The authors have selected twenty plants for further examination in silico due to their antidiabetic activity basing on literature data. The most abundant flavonoids, phenolic acids and terpenes from these plants were investigated on predicting antidiabetic action. The flavonoids were the most active for modulation of the function of 17 protein-targets and have high structural closeness to known antidiabetic preparations. The antidiabetic effects were estimated as linked with three mechanisms of action, including regulation of insulin secretion/ sensitivity; regulation of glucose metabolism; and regulation of lipid metabolism. The results are useful for understanding of antidiabetic effects of edible Mediterranean plants.

The article is well written and interesting and very appropriated to the journal.

I have only three minor recommendation: (i) to provide a figure with all structural formulae of studied substances, (ii) to short the final conclusion and introduction, namely delete of a phrase “…and to discovery lead compounds as hesperidin and rutin for the treatment of diabetes” (lines 30–31) because there is no a subject of a discovery – the antidiabetic effects of hesperidin and rutin are known and well documented in scientific literature and; (iii) to short the phrase “Thus, there is the necessity for further studies to establish the antidiabetic activity of both compounds and their derivatives” (lines 281, 282) by the same reason – the antidiabetic effects of these substances is already established and described in scientific literature.

My general opinion: this excellent article should be published after minor corrections.

Author Response

Review of the article:

Computation screening of multi-target antidiabetic properties of phytochemicals in common edible Mediterranean plants

Reviewer’s #1 comments

The article concerns the explanation of the antidiabetic action of Mediterranean edible plant materials using DIA-DB 20 inverse virtual screening web server. The authors have selected twenty plants for further examination in silico due to their antidiabetic activity basing on literature data. The most abundant flavonoids, phenolic acids and terpenes from these plants were investigated on predicting antidiabetic action. The flavonoids were the most active for modulation of the function of 17 protein-targets and have high structural closeness to known antidiabetic preparations. The antidiabetic effects were estimated as linked with three mechanisms of action, including regulation of insulin secretion/ sensitivity; regulation of glucose metabolism; and regulation of lipid metabolism. The results are useful for understanding of antidiabetic effects of edible Mediterranean plants.

The article is well written and interesting and very appropriated to the journal.

We would like to thank reviewes for the appreciative comments and constructive suggestions.

I have only three minor recommendation: (i) to provide a figure with all structural formulae of studied substances.

A new figure is added according to the reviewer’s comment.

To short the final conclusion and introduction, namely delete of a phrase “…and to discovery lead compounds as hesperidin and rutin for the treatment of diabetes” (lines 30–31) because there is no a subject of a discovery – the antidiabetic effects of hesperidin and rutin are known and well documented in scientific literature and

The phrase “…to discovery lead compounds as hesperidin and rutin for the treatment of diabetes” was eliminated. The phase “…pinpointing the most active phytoconstituents” was added.

to short the phrase “Thus, there is the necessity for further studies to establish the antidiabetic activity of both compounds and their derivatives” (lines 281, 282) by the same reason – the antidiabetic effects of these substances is already established and described in scientific literature.. 

The sentence was removed.

Reviewer 2 Report

Title: Computation screening of multi-target antidiabetic properties of phytochemicals in common edible Mediterranean plants

Herein, in silico virtual screening methodologies was used by the authors to explain the antidiabetic potential of edible Mediterranean plant materials as well as to pinpoint their most active compounds. Plants are complex mixtures of several primary and secondary metabolites and the classic approach including the isolation and the evaluation of the antidiabetic activity is time consuming and complicated. The idea is great but there are many limitations for the work

Major issues

1.      Although classic approach including the isolation and the evaluation of the antidiabetic activity is time consuming and complicated but those that showed highest should be examined in vitro and in vivo as well to consolidate the obtained results. Furthermore, instead of isolation they can be purchased

2.      Molecular docking and dynamics simulation experiments should be performed to further strengthen the results  

Minor issues

1.         The novelty of the study should be carefully addressed in the introduction section

2.         The research methodology should be comprehensively described to make it easier for readers to benefit from the work done

3.         The major structures that should pronounced activity should be drawn using chemdraw or any other software

4.         The manuscript contains some spelling, grammatical and formatting mistakes that should be revised carefully

5.         The references should be carefully checked to be all in the same style.

Reviewer 3 Report

The study is associated with assessment of anti-diabetic properties of edible plants derived from Mediterranean region. The paper is clearly written, Introduction is relevant and the aim is appropriate highlighted. The presentation of the results is clear.

I have only a few suggestions of editorial nature:

Table 1: Sometimes it is difficult to know which effects are assigned to a particular plant. Moreover, for some plants lack of the abbreviated name of the describer after the name of a species (e.g. should be  Nigella sativa L.)

Table 2. Correct the way of citation in whole Table, e.g. in first line is: “[90,91,100,101,92–99]” and should be: “[90-101].

Table 3 should be moved as close to the first mention in text as possible.

Author Response

Review of the article:

Computation screening of multi-target antidiabetic properties of phytochemicals in common edible Mediterranean plants

Reviewer’s #2 comments

The study is associated with assessment of anti-diabetic properties of edible plants derived from Mediterranean region. The paper is clearly written, Introduction is relevant and the aim is appropriate highlighted. The presentation of the results is clear.

We thank for the positive comments and the recommendations.

Table 1: Sometimes it is difficult to know which effects are assigned to a particular plant. Moreover, for some plants lack of the abbreviated name of the describer after the name of a species (e.g. should be  Nigella sativa L.)

The names of all plants were checked and corrected in the new version of manuscript.

Table 2. Correct the way of citation in whole Table, e.g. in first line is: “[90,91,100,101,92–99]” and should be: “[90-101]

The citations were checked and corrected through the manuscript.

Table 3 should be moved as close to the first mention in text as possible

The change was done.

Round 2

Reviewer 2 Report

The authors partially reply to the comments as attachment and thus the manuscript could be accepted for publication.
